# Deep learning approaches to landmark detection in tsetse wing images

**Dylan S. Geldenhuys**[1,2]*, **Shane Josias**[2,3], **Willie Brink**[2], **Mulanga Makhubele**[2], **Cang Hui**[2,4], **Pietro Landi**[2], **Jeremy Bingham**[1,2], **John Hargrove**[1,2], **Marijn C. Hazelbag**[1,5]

**1** The South African Department of Science and Innovation-National Research Foundation (DSI-NRF) South African Centre for Epidemiological Modelling and Analysis (SACEMA), Stellenbosch University, Stellenbosch, South Africa, **2** Department of Mathematical Sciences, Stellenbosch University, Stellenbosch, South Africa, **3** School for Data Science and Computational Thinking, Stellenbosch University, Stellenbosch, South Africa, **4** Mathematical Biosciences Group, African Institute for Mathematical Sciences, Muizenberg, South Africa, **5** ExploreAI (Pty) Ltd., Cape Town, South Africa

* dylangeldenhuys1@gmail.com

**Data Availability Statement:** All code used in this study is available at https://github.com/DylanGeldenhuys/Landmark-detection-for-tsetse-fly-wings. The training data as well as the final

## Abstract

Morphometric analysis of wings has been suggested for identifying and controlling isolated populations of tsetse (*Glossina spp*), vectors of human and animal trypanosomiasis in Africa. Single-wing images were captured from an extensive data set of field-collected tsetse wings of species *Glossina pallidipes* and *G. m. morsitans*. Morphometric analysis required locating 11 anatomical landmarks on each wing. The manual location of landmarks is time-consuming, prone to error, and infeasible for large data sets. We developed a two-tier method using deep learning architectures to classify images and make accurate landmark predictions. The first tier used a classification convolutional neural network to remove most wings that were missing landmarks. The second tier provided landmark coordinates for the remaining wings. We compared direct coordinate regression using a convolutional neural network and segmentation using a fully convolutional network for the second tier. For the resulting landmark predictions, we evaluate shape bias using Procrustes analysis. We pay particular attention to consistent labelling to improve model performance. For an image size of 1024 × 1280, data augmentation reduced the mean pixel distance error from 8.3 (95% confidence interval [4.4,10.3]) to 5.34 (95% confidence interval [3.0,7.0]) for the regression model. For the segmentation model, data augmentation did not alter the mean pixel distance error of 3.43 (95% confidence interval [1.9,4.4]). Segmentation had a higher computational complexity and some large outliers. Both models showed minimal shape bias. We deployed the regression model on the complete unannotated data consisting of 14,354 pairs of wing images since this model had a lower computational cost and more stable predictions than the segmentation model. The resulting landmark data set was provided for future morphometric analysis. The methods we have developed could provide a starting point to studying the wings of other insect species. All the code used in this study has been written in Python and open sourced.

landmark data can be found in the Data Dryad repository DOI:10.5061/dryad.qz612jmh1.

**Funding:** This work was supported by the Department of Science and Innovation and the National Research Foundation of South Africa. Additionally, CH and PL were supported by the National Research Foundation of South Africa, grant 89967, https://www.nrf.ac.za/. The funders had no role in study design, data collection and analysis, decision to publish, or preparation of the manuscript.

**Competing interests:** The authors have declared that no competing interests exist.

## Author summary

Tsetse flies, vectors of human and livestock disease, cost African countries billions of dollars annually. Wing shape can be used to identify isolated tsetse populations for suppression. The shape can be captured from digital images by locating anatomical points termed landmarks on the wings. Manual positioning of landmarks is prone to error and only feasible for small data sets. We analyse 14,354 wings and have developed a novel method for automatically positioning landmarks on insect wings. We use a two-step approach, first removing wings that are missing landmarks due to damage. We thereby provide accurate landmarks, unbiased for wing shape. We compared two modern deep-learning models to locate landmarks and showed how data manipulation techniques improve performance. We deployed the final system to generate the landmark data for the tsetse wing images and provide the trained models for transfer learning on similar tasks with smaller data sets. Our methods will allow researchers to study tsetse wing shape in relation to other biological data and to gauge how wing shape characterises tsetse populations. Our method could provide a starting point to the study of the wings of other insect species.

## Introduction

There has been a growing interest, for more than two decades, in the use of morphometric analysis in medical entomology [1] including the study of tsetse flies (*Glossina spp*) and their control. Tsetse are the vectors of human and animal trypanosomiasis in Africa, the latter disease costing 37 African countries USD 4.5 billion annually [2]. There has been particular interest in the morphometry of tsetse wings, on which the relative positions of 11 landmarks Fig 1 are used to assess the wing size and shape. Morphometry provides a useful taxonomic tool [3] and, of practical significance in terms of disease control, a simple inexpensive method for identifying isolated populations of tsetse. In the latter role, wing shape is used as a phenotypic expression of the underlying genetic signature of the population [4–11]. It is argued that, where wing shapes differ significantly between populations, this may be useful in identifying sufficiently isolated populations that can be targeted for local eradication without risk of reinvasion from neighbouring populations.

The utility of morphometric analysis in identifying isolated populations depends on the extent to which the wing shape and size variation depends on environmental factors. For example, suppose wing morphometry changes substantially with climate and season within a single isolated population. One would then need to decide whether observed differences between the wing morphometry of flies sampled from two populations reflect differences between the populations—or temporal, environmentally induced, changes in either or both populations. To date, morphometric studies of tsetse have not looked for temporal or seasonal variation because sample sizes have been too small. Almost all studies have fewer than 200 individual flies of any given sex and species, sampled from a given locality and generally at a given time. As such, they would be of limited use in assessing the importance of temporal changes. The study of Mbewe et al [11] is an exception, having sample sizes of about 1000 wings from each sex of *G. f. fuscipes* sampled from four islands in Lake Victoria, western Kenya, before and after a trapping-out exercise. However, this study was more concerned with the effects of the trapping on wing size and shape rather than any potential temporal or seasonal effect.

All of the studies quoted above were based on sufficiently small numbers of wings that it was feasible to locate the wing landmarks manually. In the present study, we use wings from a

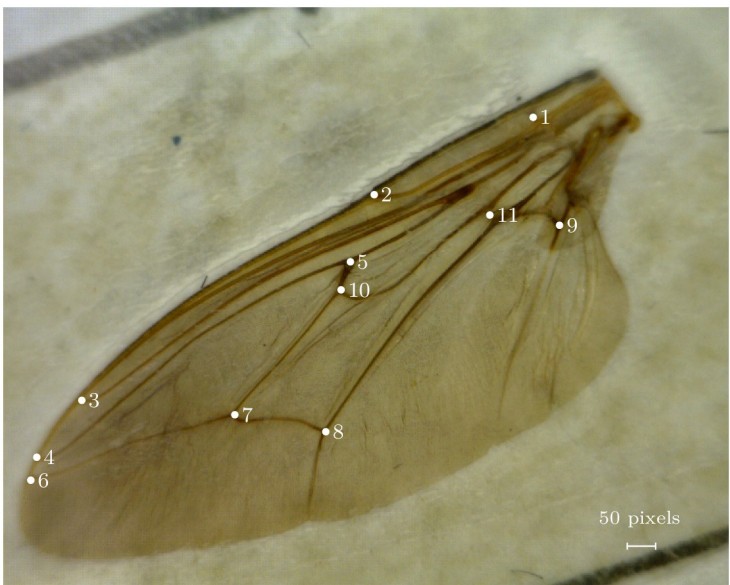

**Fig 1. Image of a tsetse wing containing 11 landmarks indicated by white numbered points.** The image also contains a scale that can be useful for placing later errors into context. Each pixel equates to approximately 0.007mm based on images taken of a physical measure.

study where the sample size is orders of magnitude greater—and, therefore, it was essential to develop an automated method for assigning landmarks. Between September 1988 and December 1999, more than 200,000 tsetse (*G. pallidipes* and *G. m. morsitans*) were sampled within 2 km of a single site, Rekomitjie Research Station, in the Zambezi Valley of Zimbabwe. Females were subjected to ovarian dissection to determine their age [12], and subsets were analysed for nutritional status and infection with *Trypanosoma vivax* and *T. congolense*. During the study, the wings of each fly processed were fixed with transparent adhesive tape to the results forms, which were later laminated between transparent plastic sheets to protect the wings.

Over the past 30 years, subsets of the data have been used in numerous publications [13–34]; these provide, among other things, full details of dissection and analytic procedures. As part of the original study, the length of each wing was measured using a binocular microscope. These lengths have been used in several of the publications listed above—most pertinently in an analysis of the association between wing length and sampling method [31]. In the current work, we put the data to an entirely new purpose, making morphometric measurements on the wings of a large subset of the flies processed.

We are primarily concerned with solving the technical problem of developing a semi-automated system that assigns Cartesian coordinates to the 11 landmarks for each tsetse wing. Each of these is defined by points of wing vein intersection (Fig 1), which have been used in previous morphometric studies on tsetse [3–11]. To limit the effort required for data cleaning, we used a subset of all available data, consisting of 14,354 pairs of wings from tsetse captured over nearly two years in every season and month of the year. The results should thus reflect much of the seasonal variation found in the entire data set. The selection also provides a sufficiently large sample to evaluate our attempts at developing an automatic landmark detection system.

The image data set contains wings missing some landmarks due to damage occurring during fly capturing and processing. We define wing images containing all landmarks as complete

wing images and refer to those missing landmarks as incomplete. We developed methods for detecting and removing incomplete wing images from the data before performing automatic landmark detection for the remaining complete wings.

To our knowledge, there have been no published studies addressing the problem of identifying damaged wings or the challenge of automatic landmark detection in tsetse wing images. With similar wing images, deep learning models have been applied successfully in classifying fruit fly species [35]. Although detecting incomplete wings was not of concern in this study, their methods may be a good candidate for our task since it also aims at wing classification.

There have been numerous studies attempting landmark detection on other insect wings, most notably on the wings of the fruit fly, genus *Drosophila*. Contrary to the data set in the current study, the data sets in these studies are relatively small. Consequently, the authors used manually chosen features and a machine learning approach to reduce the amount of data needed for model training [36–38].

Convolutional neural networks (CNN) architectures are often used on large data sets to automatically learn the best bottleneck features for both classification and regression tasks. This is due to their ability, given sufficient data, to capture complex nonlinear relationships between input images and the desired output. In facial landmark detection and biomedical imaging, deep CNN architectures have been used to locate landmarks accurately. Current state-of-the-art models in facial landmark detection are direct coordinate regression using CNNs or heat-map regression using fully convolutional networks (FCN) [39]. Both of these techniques are also currently used for landmark detection in biomedical image data [40–42]. FCNs are often used for segmentation tasks but are adapted for heat map regression when used for landmark detection. Previous research has not studied segmentation for landmark detection in images, perhaps due to disproportionately more background pixels than landmark pixels. This class imbalance problem is well-known in segmentation tasks but has been overcome in recent years following the development of new loss functions [43–45].

The models mentioned above use supervised training, which relies on labelled data. Recent papers on data-centric approaches note that model performance can improve significantly by focusing on consistent and accurate data labelling for training, as well as data augmentations [46–48].

This paper uses a two-tier deep learning approach to classify tsetse wing images as complete or incomplete, followed by landmark detection. We compare the generic state of the art computer vision models for classification for the first tier. For the second tier, we compare direct coordinate regression using a CNN and segmentation using a FCN called UNet++ using dice loss [49–51] to combat the large class imbalance between background (non landmark) and foreground (landmark) pixels. The segmentation task is defined similarly to that used by Vandaele et al. [36] in that pixels in a given radius around a particular landmark are segmented from the rest of the image, averaging their locations to determine the final landmark location. These models are also compared with and without data augmentation. We employ a data-centric approach to develop these models, sampling training data from the data set of interest and accurately annotating these data for classification and landmark detection. We evaluate the effect of prediction errors on subsequent morphometric analysis and apply the models to the full un-annotated data set. To validate the classifier's performance on the un-annotated data set, we compare the expected proportion of classifications inferred from sample statistics to the predicted proportion. Finally, we perform analysis for checking data alignment between the biological and image data set with predicted landmarks. We correct and remove identified errors in the data set and provide the final landmark data set for subsequent morphometric studies. The workflow described is illustrated in Fig 2.

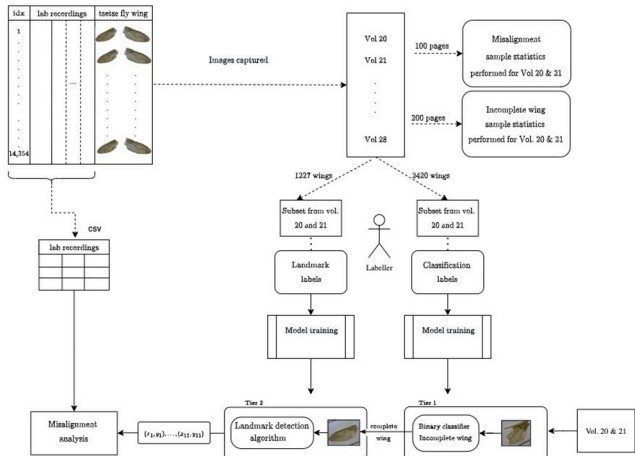

**Fig 2. Flow chart of the entire procedure used in this study, including the tier 1 and 2 development and deployment.** The data were first recorded on paper and laminated with the physical tsetse wings, separated into volumes consisting of pages illustrated on the top left of the diagram. Biological lab recordings and tsetse wings were then digitally captured in a CSV file and nested folders (i.e. Vol/page) of images. This study used a subset of the full data set (Vol. 20 and 21) to establish a method for recording landmarks automatically. *Labeller* refers to the manual labelling stage to train machine learning models. Sample statistics were performed to understand the proportion of different categories of incomplete wings, which was used to inform an appropriate classification model. In addition, sample statistics were performed on misaligned pages to estimate the number of misalignment pages we expect to find. The tier 1 and 2 processes at the bottom of the figure explain the deployment process. Tier 1 decides whether a wing is complete and can be sent to tier 2, where landmarks are localised. The two-tier landmark detection system is deployed on the unannotated data set of all images in Volumes 20 and 21. The final *Misalignment analysis* is fully described in Fig A of S1 Fig.

## Materials and methods

### Data description

More than 201,545 tsetses (*Glossina pallidipes* and *G. m. morsitans*) were collected in an 11-year study carried out at Rekomitjie Research Station in the Zambezi Valley of Zimbabwe. Subsets of the flies were subjected to nutritional analysis and, for females, to ovarian dissection to determine their age. All biological variables and descriptions can be found in Table A in S1 Text. The features used in this study for analytical purposes are given in Table 1.

All details relating to each collected fly were recorded on a single line of an A4 sheet of paper, and the tsetse wings were fixed on this line with transparent adhesive tape. A standard measure of wing length was determined for one wing of each pair, using a binocular microscope fitted with a graduated reticle in one eyepiece. The distance between landmarks 1 and 6 (refer to Fig 1) was used as this standard measure, converted to a length (*wlm*) in millimetres by allowing for magnification differences between microscopes. Each completed page was laminated between transparent plastic sheets to protect the wings further. Each page contains a

**Table 1. Biological data captured in lab dissection.**

| Variable name | Description |
| --- | --- |
| vpn | Volume, page and number |
| wlm | Wing length measured from landmark 1 to 6 (mm) |
| lmkl | Number of missing landmarks for left wing |
| lmkr | Number of missing landmarks for right wing |

maximum of 20 wing pairs. The pages were collected in 28 volumes. The wings from volumes 13 to 25 have been digitized to $1024 \times 1280$ resolution images using a high-resolution microscope camera. This study only considers volumes 20 and 21, which refer to flies collected in 1994 and 1995. The biological data from these volumes have been digitally recorded, and quality checked for morphometric studies. Volumes 20 and 21 comprise 14,931 pairs of wings, which is only 7.4% of the 201,545 pairs collected between September 1988 and December 1999. Fig A of S2 Fig indicates, however, that the variation in wing length in volumes 20 and 21 covers a majority of the seasonal, and overall, variation observed in the parent set. As such it seems reasonable to think that the subset is sufficiently representative of the whole to support an initial attempt at developing an automatic landmark detection system.

The image data are kept in a folder structure matching the physical tsetse data set. Each image is named according to the location of the physical fly wing; volume, page, line, and left- or right-wing. For example, 'V20P076L08R' refers to volume 20, page 76, line 8, right-wing. The name of each wing image links it to the biological recordings. It is important to note that errors due to photographing the incorrect wing or incorrectly naming the page would result in misalignment between the biological and image data. Misalignment may extend to multiple wings on a page due to a single error; for example, were the photographer missed a particular wing, the upcoming sequence of photographed wings on the same page would be misaligned. Such misalignments have been noted and addressed in this study.

To create a suitable data set for geometric morphometric analyses, we require all landmarks to be present consistently. We only considered making landmark predictions on complete wings where all 11 landmarks are visible. The types of incomplete wings fall into various categories, including missing wings, tears, ink stains covering landmarks, and other missing pieces of the wing. Examples of such incomplete wings can be found in Fig A of S3 Fig. Within the class of complete wings, we also noticed a small proportion of wings with defects such as tears, ink stains, and some deformation.

The complete wings primarily vary in size, rotation, and location. Images also differ in colour contrast and brightness. Prior sample statistics were performed from a single random sample of 200 wings to understand the proportion of incomplete wings in the data. The sample statistics, reported in the results section, showed that the majority of incomplete wings are missing landmarks 4 or 6.

In our initial efforts, we attempted to remove wings using the *lmkr* and *lmkl* attributes in the biological data set, which indicate how many landmarks are missing in the right- and left-wing. However, visual inspection of the images accompanying these attributes revealed that the attributes were sometimes labelled incorrectly. Mislabelling was likely due to either misalignment between images and biological data or incorrect recording of missing landmarks. Therefore, we could not rely on these attributes to remove all incomplete wings.

## Workflow for landmark detection

For simplicity, we flipped all right-wing images horizontally to obtain a data set of only left-wing images. We then removed incomplete wings using a classification approach. Since most incomplete wings are missing landmark 4 or 6, we trained a classifier to identify wings missing these landmarks. We benchmarked multiple state-of-the-art deep learning models on our data set to obtain a suitable classifier. We then trained deep neural networks to produce landmark coordinates on complete wings. We performed image augmentations for the first tier training and experimented with and without augmentations in the second tier. To evaluate the models, we used a single independent train-validation-test split. To examine the potential adverse effects of the landmark prediction errors on subsequent morphometric studies, we determined

whether varying shapes correlated with prediction errors by analysing the impact of the geometric shape disparity on the average pixel distance errors. The models were then applied to all images from volumes 20 and 21. To ensure the landmark predictions and biological data were aligned correctly, we calculated the correlation between measured wing length (*wlm*) and the predicted wing length for each page. We expected a linear relationship with a high correlation between the wing lengths for well-aligned pages. Hence we identified pages with low $R^2$ values to be inspected for misalignments. The identified misalignments were corrected or removed from the data set. Finally, the measured wing length (*wlm*) and predicted wing length were plotted for all pages and examined. The standard error was used to measure the agreement between the predicted and manually measured wing length. We visually examined the inliers (within prediction error interval) to inspect the predictions. Outliers were also examined for mistakes such as remaining misalignments and incorrect incomplete wing classifications. These instances were removed from the final data set.

## Incomplete wing classifier

**Training data.**   We curated a data set consisting of a class of complete wing images containing all landmarks and a second class consisting of incomplete wing images missing landmarks 4 or 6. Since manually finding training samples is time consuming, we aided the process by using the biological data set (variables *lmkr* and *lmkl*) and then used visual inspection to filter a set of training images. A data-centric approach was employed, focusing on consistency in labelling, such that the types of wings we introduced into the training set were not ambiguous and fell into well-defined groups. We removed examples that did not clearly fall into either class. The resulting data set consists of an even class distribution of incomplete and complete wing images. Images are labelled 1 for incomplete wings and 0 for complete wings. In total, we labelled 1227 images.

**Convolutional neural network classification models.**   We compared three modern computer vision models with transfer learning that were used similarly for classifying insect wings in a previous study [35]. These models are ResNet18 [52], Inception [53], and VGG16 [54] with batch normalisation. We replaced the final fully connected layers, after the convolutional layers, with a fully connected layer of size 1. We then applied a sigmoid activation function, with an output in the range (0, 1).

A 3-channel input was used for all models. Input image size of $244 \times 244$ was used for both VGG16 and ResNet18. For the Inception model, we used an input image size of $299 \times 299$ since the default kernel size of $3 \times 3$ and a stride of 2 with zero paddings require the image dimensions to be odd numbers. We fine-tuned each classifier with unfrozen weights for 30 epochs with a batch size of 50. The model was saved at the lowest validation score in each training session. A learning rate of 0.0001 was used with the Adam optimiser and a binary cross-entropy loss function.

**Evaluation**: For each classifier, we obtained 95% confidence intervals for specificity, sensitivity, precision, f1 score, and accuracy by bootstrapping the predictions on the test set, randomly sampling 205 samples, 5000 times to create sample statistic distributions.

## Landmark detection model

**Landmark data.**   To train the landmark detection models, we used a data set of 2420 unique complete wing images (one wing per tsetse fly) sampled from volume 20 and 21, for which landmarks were precisely annotated by a single person and subsequently reviewed by others. A custom user interface was used to annotate the images digitally.

**Evaluation**: All model outputs were scaled to match the original image size and were evaluated in terms of pixel distances, using mean absolute error (MAE) and root mean squared error (RMSE). Test set MAE is reported for baseline model obtained by calculating the mean location for each landmark. In addition, we compared the performance of each model with and without data augmentation.

**Convolutional neural network regression model.** We first frame landmark detection as a regression problem, i.e. we directly predict the $(x, y)$ coordinates for the landmarks. For this task, we utilised a ResNet50 network with weights pre-trained on ImageNet [55]. ResNet tends to favour a smooth loss landscape, allowing for a larger architecture while maintaining stable optimisation [52]. We altered the architecture by adding a randomly initialised convolutional layer, followed by a fully connected layer with 22 outputs corresponding to each $x$ and $y$ coordinate for the 11 landmarks. Fig 3 provides a high-level illustration of the model.

We used mean square error (MSE) loss with the Adam optimisation function to train the model. Two training sessions were conducted; each session ran for 100 epochs, initiated with a learning rate of 0.001 and reduced to 0.0001 after the first session. The model was saved at the lowest validation score in each training session.

**Fully convolutional network segmentation model.** As an alternative to the regression problem framing, we considered landmark detection as a semantic segmentation problem. Semantic segmentation is considered a dense prediction task, where the output is an activation map with each pixel being assigned a label. For landmark detection in tsetse wing images, this becomes a supervised learning problem where a label is a binary segmentation map for each landmark, with a disk centred at the $(x, y)$ position of the landmark, as shown in Fig 4. The dimension of the segmentation map is the same as the input dimension.

For the segmentation model we chose the UNet++ [56] architecture, using the implementation as in [57]. UNet++ is based on the fully convolutional UNet [58] architecture that is divided into two blocks. The first is a downsampling block, responsible for capturing global context and low-frequency content of which objects of interest are usually comprised. The second is an upsampling block, responsible for precise localisation of objects [58]. UNet++ extends UNet by including skip-pathways between layers in the downsampling block and the

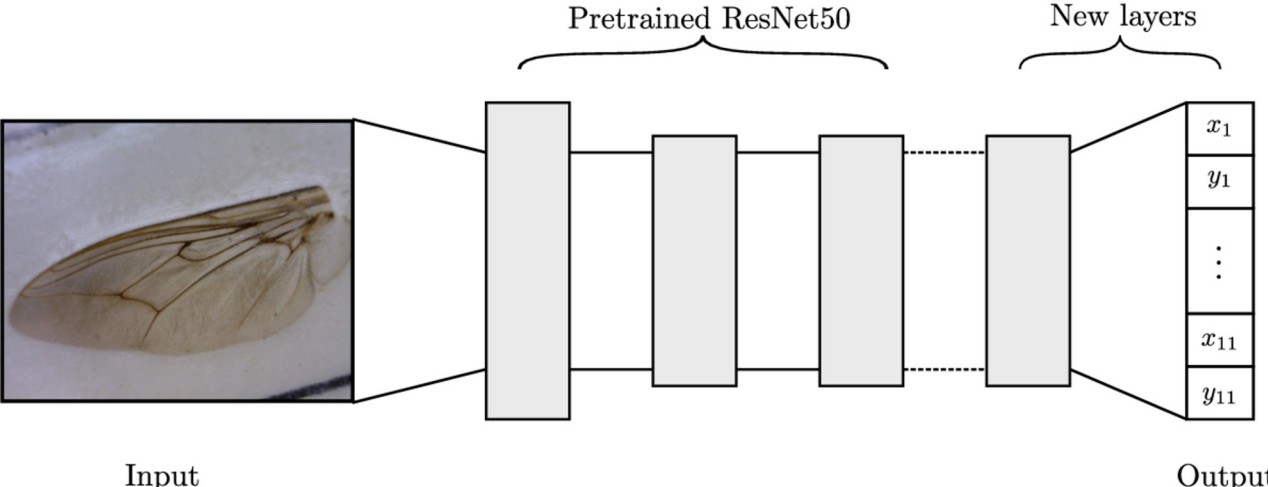

**Fig 3. ResNet50 is modified by removing the final 2 layers and replacing them with a randomly initialised convolutional layer, followed by a fully connected layer of size 22, representing the output.** The output corresponds to an $x$ and $y$ coordinate for each of the 11 landmarks.

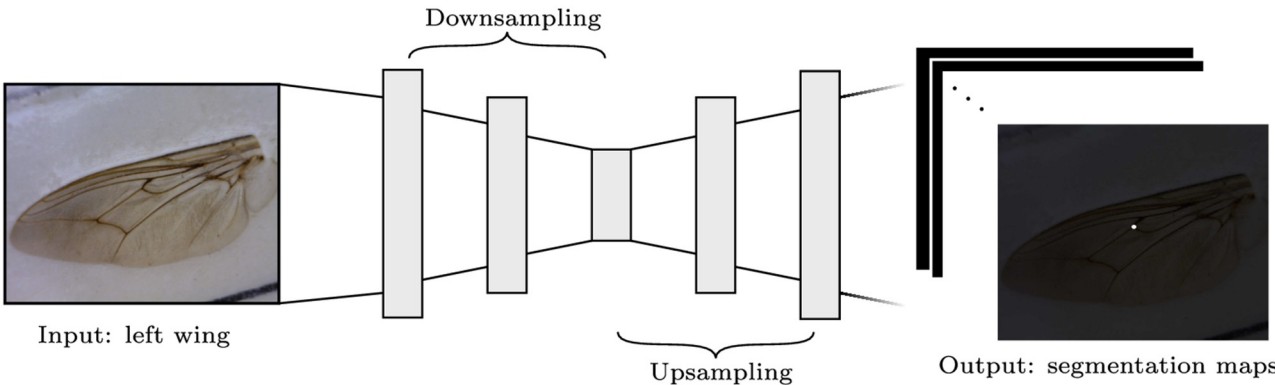

**Fig 4. The network is composed entirely of convolutional layers.** It can be divided into downsampling and symmetric upsampling blocks. The output is of dimension $11 \times 224 \times 224$, where each output segmentation map is a binary image with a disk centred at a particular landmark.

upsampling block so that the semantic gap between feature maps in those layers is reduced. Zhou et al. [56] argue that reducing the semantic gap leads to an easier optimisation problem. A high-level illustration of the architecture is shown in Fig 4.

We took the average between a binary cross-entropy loss and dice loss applied to the outputs to train the model. We trained the model from scratch over 50 epochs, using the Adam optimisation function with a learning rate of 0.001. The model was saved at the lowest validation score in each training session.

To obtain landmarks from the segmentation maps, we determined the average location of all pixels greater than or equal to the seventh-highest pixel value in the map. This inference process produces a landmark, even if it is absent (as does the regression network).

### Training implementation details

We employed transfer learning for the first tier and the regression model in the second tier [59], using networks pre-trained on ImageNet [55]. In addition, we used image augmentations during all first-tier training and experimented with and without augmentations for the second tier training to improve model robustness. The augmentations increased the noise and size of the training data, preventing over-fitting and ensuring that the training data contained images on the extreme of the spectrum for wings available in volumes 20 and 21, in terms of wing location, rotation, and size which comprised most of the noisy characteristics of the data. These data augmentations helped maintain a high signal-to-noise ratio in the above mentioned noisy cases. We randomly sampled from a [−5%, 5%] interval for scaling and shifting, and a [−22°, 22°] interval for rotation to transform each image. We chose these intervals based on a visual inspection of what degree of augmentation seemed reasonable. We use a single cross-fold validation strategy for all model training with a 60:20:20 train, validate, and test split.

### Hardware and software

The code was run on a GTX 1650 4GB GPU with a 4GHz i7 CPU and 8GB RAM. All of the code was written in Python version 3.7. The pre-trained models were taken from the Pytorch Computer Vision Library and altered for our purposes. The code can be found in a GitHub repository [60]. The data can be found on Dryad [61].

### Effect of prediction errors on subsequent morphometric analysis

To assess the effect of prediction errors on subsequent geometric morphometrics, we evaluated the effect of the Procrustes shape disparity on the average prediction error in a wing image. In particular, we plotted the Procrustes disparity (Procrustes distance from the average shape) as the independent variable and mean pixel distance error as the dependent variable. We then fitted a regression line to determine the correlation and linear relationship.

It is necessary that the model only infers the landmark position and not the wing shape since it is the landmarks that determine the wing shape, and it will be difficult to determine whether the model is inferring the correct wing shape irrespective of the landmark error. If a relationship exist between the error and shape of the wing, then there is evidence that the model is performing better on some wing shapes than others making it impossible to compare different wing shapes in later morphometric analysis.

### Data alignment and correction

In this study, we faced the problem of having two data sets that were not perfectly aligned. That is to say, some wings labelled with the same identifier in the two data sets did not refer to the same wing.

The approach for finding misalignments is outlined as follows. We estimated the proportion of pages containing misaligned data by generating a random sample of 100 pages and manually checking these pages for misalignments. From this we calculate an error estimate using a 95% confidence interval. We performed landmark predictions for each wing and calculated the $R^2$ value for each page of wings using measured wing length *wlm* vs wing length between predicted landmarks 1 and 6. We expected pages with very low $R^2$ values to indicate wings not aligning with the correct *wlm* measurement. We chose a $R^2$ threshold of 0.1, which made the percentage of pages indicated as misaligned slightly higher than the upper bound of the confidence interval for the estimated percentage of pages misaligned. We then manually checked these pages for misaligned data. Once the pages with misaligned data were found, we considered whether we had successfully corrected or removed the misaligned data by comparing the number of pages found with the proportion estimated in the random sample.

## Dryad DOI

https://doi.org/10.5061/dryad.qz612jmh1.

## Results

### Incomplete wing classifier

The sample statistics showed that 13% ± 4.63% of wings are incomplete. The majority of incomplete wings were missing landmark 4 or 6, making up 10% ± 4.13% of the data, comprising on average 77% of all incomplete wings. We use an online sample size calculator [62] for these sample statistic calculations.

After evaluating each trained model on the test set and generating a 95% confidence interval for each model, we found VGG16 with batch normalisation to have the best performance, achieving a perfect score for all metrics. ResNet18 and Inception had a trade-off between sensitivity and specificity, with ResNet18 having a perfect specificity and Inception having a perfect sensitivity. The training loss curves can be found in Fig A in S2 Text, and the scores for each model can be found in Table A in S2 Text, Table B in S2 Text, and Table C in S2 Text.

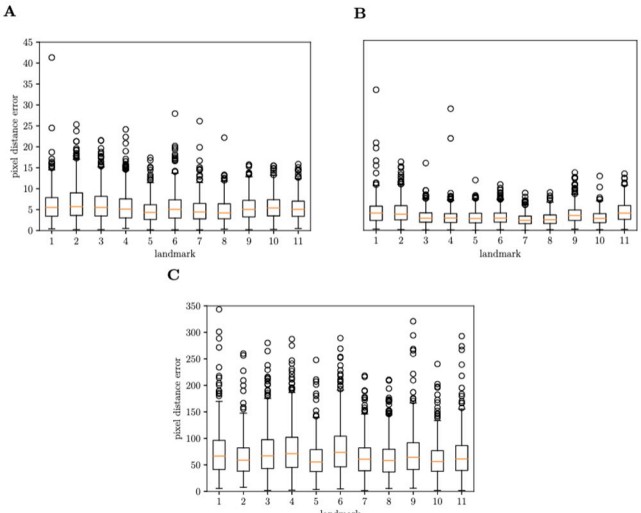

**Fig 5. Box-and-whisker plots for the baseline model, regression and segmentation networks.** The regression and segmentation network show significant improvement over the baseline (5C). The regression network (5A), has a slightly higher mean pixel distance error and higher maxima when compared to the segmentation network, but has fewer egregious outliers. For the segmentation network (5B), four outliers ranging from 50 to 570 are not displayed for clarity.

## Landmark detection model

The Euclidean pixel distance errors are given as a box plot in Fig 5. For all 11 landmarks, we obtained a mean average error (MAE) of the Euclidean pixel distance of 5.34 (95% confidence interval [3,7]) for the regression network and 3.43 (95% confidence interval [1.9,4.4]) for the segmentation network. Both the regression and segmentation network show significant improvements over the baseline (mean location per landmark), further motivating our methods. The performance of the baseline can be explained by the variance in orientation and position of the wings in the images. For root mean squared error (RMSE) we obtained 6.32 for the regression network and 6.67 for the segmentation network on the original image size ($1024 \times 1280$). Since we observed skewed error distributions, we further presented the box plot of the first and third quartiles for the spread of errors. It is worth noting that there are variations among landmarks in their error distributions.

When running the models on the test set the regression network was approximately 5 times faster than the segmentation network.

Table 2 shows the average landmark errors for each network with and without augmentations. The segmentation network did not benefit from image augmentation but the regression network improved significantly.

**Table 2. Effects of data augmentations on average landmark errors.**

|  | Segmentation network | | | Regression network | | |
|---|---|---|---|---|---|---|
|  | **LB** | **Median** | **UB** | **LB** | **Median** | **UB** |
| Without augmentations | 2.0 | 3.1 | 4.6 | 4.4 | 8.3 | 10.3 |
| With augmentations | 1.9 | 3.0 | 4.4 | 3.0 | 4.8 | 7.0 |

LB and UB refer to the upper and lower bounds of the 95% confidence interval for pixel distance error.

## Effect of prediction errors on subsequent morphometric analysis

Fig 6 shows the relationship between the mean pixel distance error, and Procrustes shape disparity from the mean shape, after removing outliers that are more than 2 standard deviations from the mean. We remove outliers for linear regression since the method used for linear regression is sensitive to outliers. The best fit regression line shows a positive slope with an $R^2$ value of 0.09 (0.16 before removing outliers) for the regression network, and 0.01 (0.12 before removing outliers) for the segmentation network.

The correlation is largely due to a proportion of outliers above and below two standard deviations from the mean; 7.2% of test data points were outliers for the regression network and 3.7% for the segmentation network. Fig 6 also shows how the average spread of errors is wider and slightly more correlated with Procrustes disparity than the segmentation model, but does not contain extreme outliers which indicates that small changes in the input will not result in a surprisingly large error, hence the regression model may be considered more stable. Examples of inlier predictions plotted with the ground truth are given in Fig A of S4 Fig.

## Application to volumes 20 and 21

We applied the missing landmark classifier to volumes 20 and 21 consisting of 14,354 pairs of wing images. We removed all cases where the corresponding image name was missing, and the number of incomplete wings classified was 2,299 out of a total of 28,708 wings (8%). The proportion of incomplete wings removed agrees with the proportion estimated from the random sample. Visual inspection of predictions indicates that the classifier can discriminate accurately between classes, often classifying more categories of incomplete wings as long as landmarks 4 or 6 were missing.

For the remaining complete wing images we decided to deploy the regression landmark model due to its lower computational cost and more stable results with regard to producing extreme outliers. The regression model also shows correlation between the errors of each landmark which helps to identify outliers (damaged wings missed by the classifier) by comparing the wing lengths of the lab recorded wing length (*wlm*) and calculated wing length from predictions. The segmentation may still produce very accurate predictions for calculating wing length while others can have a very large error as can be noted in Fig 5B.

**A**          **B**

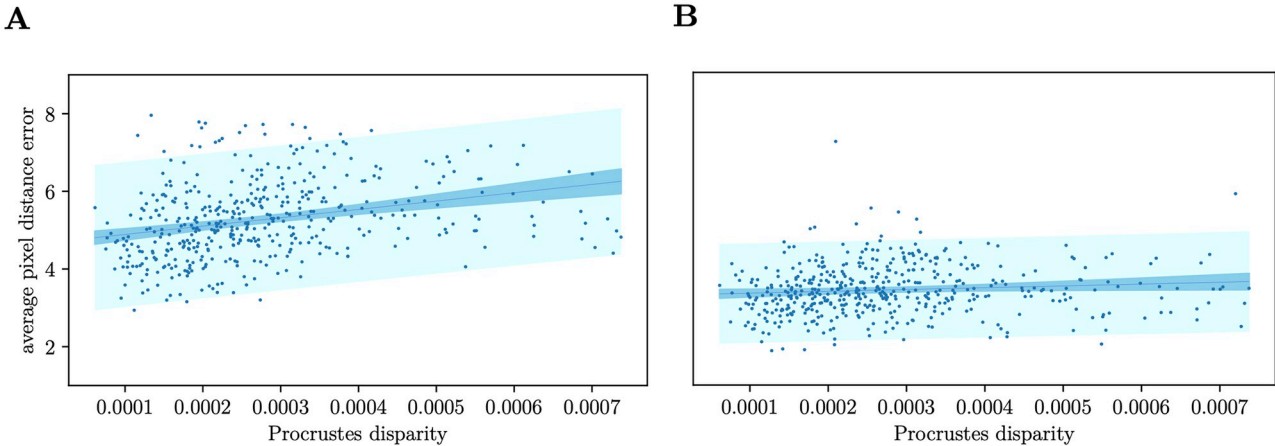

**Fig 6. Procrustes disparity from the mean landmark shape and predicted vs the mean pixel distance error.** The predictions using the regression network (A) has slightly higher correlation and pixel distance error then the segmentation network (B). The segmentation network also has a lower error interval indicated in light cyan, the 95% confidence interval is indicated in sky blue and the best fit line in royal blue.

**Data alignment and correction.** We estimated 2% ± 2.56% of pages to be misaligned based on a random sample of 100 pages. That is, we would expect around 15 of the 770 pages to be misaligned. Using our semi-automated procedure, we found 15 misaligned pages using the 0.1 $R^2$ threshold, including the two misaligned pages found in the random sample. The relevant information regarding the data cleaning can be found in Table A in S3 Text, Table B in S3 Text, under the sections named *Sample statistics for misaligned pages* and *Alterations to volume 20 and 21* in S3 Text.

**Measured vs predicted wing length.** Fig 7 shows the relationship between measured wing length (*wlm*) and the predicted wing length measured between landmark 1 and 6 after correcting the misaligned data. We observed a strong linear relationship between the two variables with an $R^2$ of 0.83. It is noted in appendix B that the *wlm* variable was measured for the right wing; otherwise, the hatchet cell was measured (distance between landmarks 7 and 11). We excluded all these cases in Fig 7. The percentage of outliers with errors larger than the standard prediction error (shown in light blue in Fig 7) was approximately 2.2%. The majority of outlying wings, however, received good predictions, which may indicate errors in the dissector's measurements or remaining misalignments (see an example in Fig 8I). Other outliers represent damaged wings or missing wings that were not detected by the classifier (Fig 8). These outliers were expected since the classifier only identifies wings missing landmark 4 or 6. The distant outliers at the bottom right of Fig 7 are shown in Fig 8G and 8H. These were all images with missing wings or wings that were misaligned. Fig 8G represents a misaligned wing where the direction of the wing and image name are inconsistent. The outliers showcased in Fig 8 were manually removed from the final landmark data set.

Besides examining the outliers, we also explored potential errors and the overall quality of predictions amongst the inliers. These make up the majority of wings that will be used for geometric morphometric studies. Most inlier predictions were of good quality and had accurate landmarks, with a few exceptions. Fig 9 shows some of the inliers with relatively large errors compared to the average prediction accuracy errors, or good predictions but for deformed wings that have altered the wing shape.

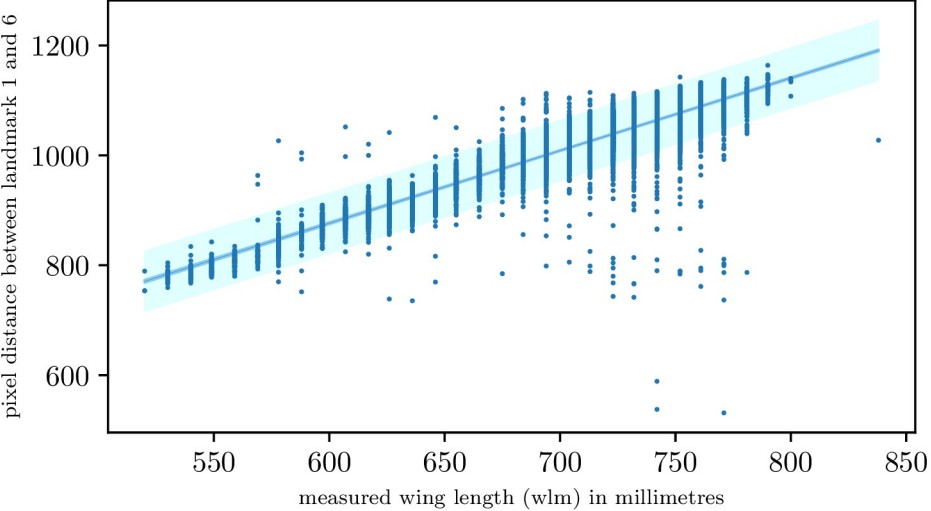

**Fig 7. Relationship between wing length measured by the dissector vs wing length calculated from the predicted landmarks 1 and 6.** The light cyan indicates the prediction interval. The 95% confidence interval (sky blue) is indistinguishable from the line of best fit (royal blue).

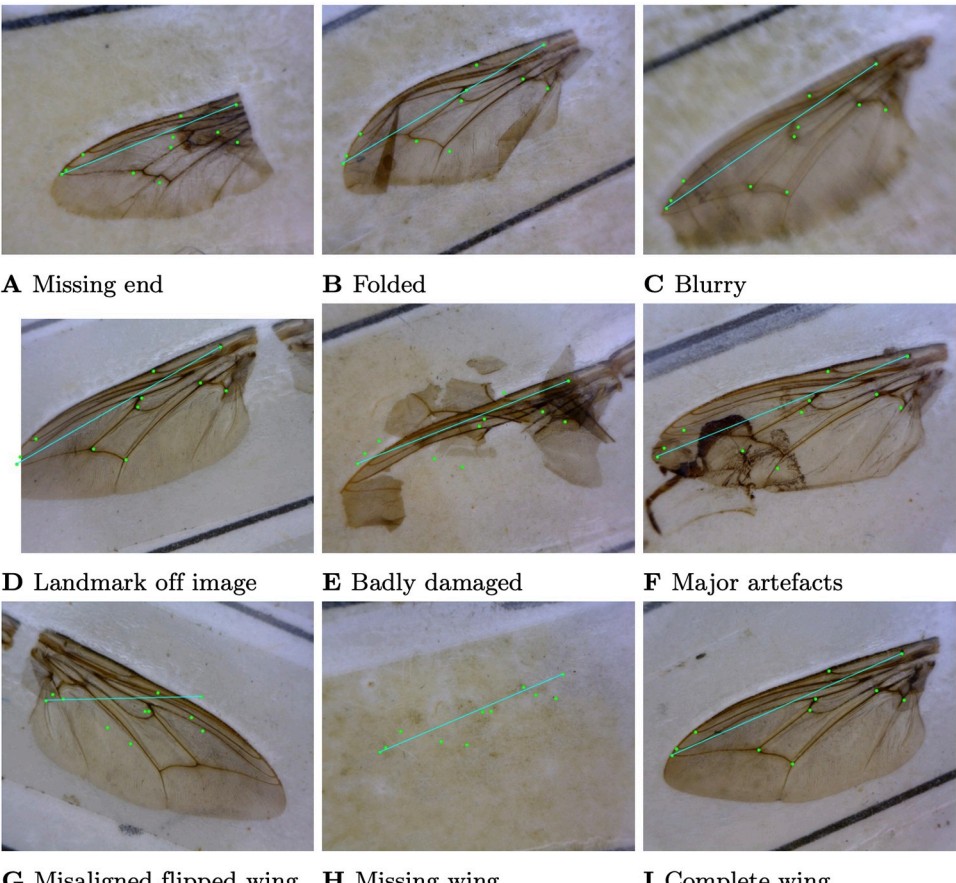

**A** Missing end    **B** Folded    **C** Blurry

**D** Landmark off image    **E** Badly damaged    **F** Major artefacts

**G** Misaligned flipped wing    **H** Missing wing    **I** Complete wing

**Fig 8. Wing images corresponding to outliers.** Dots represent predicted landmarks from the regression network and the straight line indicates wing length.

## Discussion and conclusion

Using a two-tier deep learning approach, we accurately filtered out damaged tsetse wings that were missing landmarks and provided precise landmark coordinates for the remaining wings. We showed that wing shape characteristics had a minimal effect on the accuracy of landmark predictions. In addition, we addressed the misalignment problem, ensuring that the vast majority of links between the resulting coordinates and the field-collected biological data are correct. The result is a landmark data set with biological data for morphometric analysis.

Concerning the classification task, we used a technique similar to that used by Leonardo et al. [35] for classifying fruit fly species. Leonardo et al. [35] also achieved the best results using the VGG16 network compared to ResNet50, Inception and VGG19.

Concerning the data alignment problem, other approaches used the similarity of several quasi-identifiers (QIDs) to align data [63–66]. The QID refers to some variable contained in both data sets that can be used to link the data or find misalignments in the data based on their similarity. This approach would be inadequate for our case since we only have one QID (wing length) available. Moreover, dissimilarities in QID value between the data sets may be due to factors other than misalignments. We were able to find misalignments successfully using a page-level correlation approach with wing length as a QID.

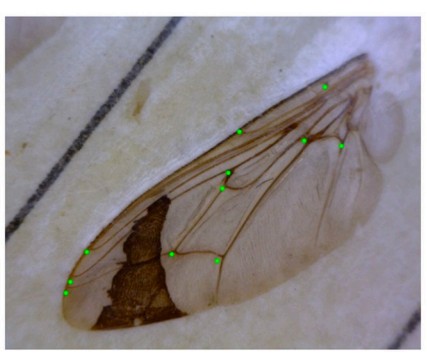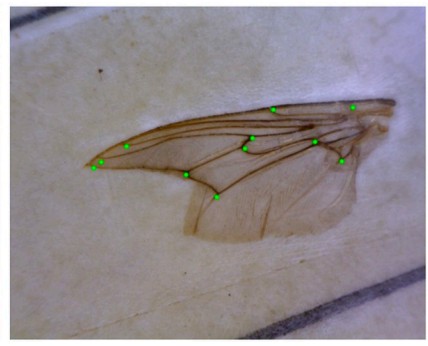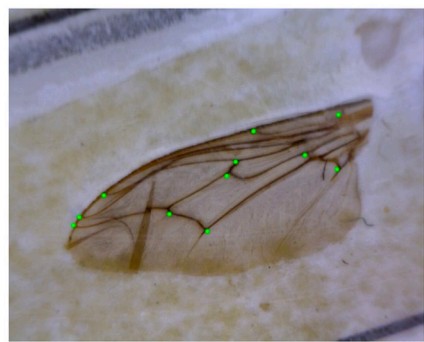

**A** Ink stain     **B** Missing piece     **C** Folded

**Fig 9. Examples of erroneous inliers.** Dots represent predicted landmarks from the regression network.

The results for the landmark model comparison indicate that the segmentation model performs slightly better than the regression model concerning pixel distance error and wing shape bias. The Procrustes analysis suggests that the segmentation model is slightly less affected by extreme changes in wing shape. However, the correlation between prediction error and the Procrustes disparity is minimal for both models.

The segmentation model contains around 1.3M more parameters, which amounts to a 3% increase in the number of parameters compared to the regression network. However, the increased model complexity is justified by the increased task complexity created by the larger output size ($11 \times 224 \times 224$), compared to the regression output ($11 \times 2$). The larger size of the segmentation model comes at an extra computational cost, taking almost five times longer to produce predictions using a GPU while also requiring more GPU VRAM. The segmentation model has high accuracy but produces extreme outliers, which the regression model does not. This implies that small perturbations in the input space can cause major errors in some landmark predictions which we refer to as model instability. The regression model uses an MSE loss function that heavily penalises distant outliers, hence producing more consistent outputs with fewer outliers. The regression model could also benefit from existing loss functions and regularisation, which have been shown to improve landmark detection [67]. In addition, it also showed a significant improvement using data augmentations as opposed to the segmentation model in which the effects of data augmentations were insignificant.

Various studies have manually identified landmarks on tsetse wings [3, 5, 10]. To our knowledge, however, this study is the first to use automatic landmark detection on tsetse wing images. We employed a data-centric approach, obtaining a large sample of quality checked training images with no incorrect labels and accurate landmark annotations. In addition, we improved the performance of the regression model using data augmentations to increase the variety of wings concerning positions, sizes, and rotations. These augmentations prevent the model from learning noisy patterns or biases in wing position, size and rotation [46]. Another strength of the current study is that we analysed how the prediction errors affect subsequent morphometric analysis, showing that our approach results in a model with only slight wing shape bias. Bias towards specific shapes would have limited the potential for morphometric studies because wing shapes at the tail of the shape distribution will have less accurate landmark predictions for determining the wing shape, consequently affecting variation and co-variation of wing shape.

A key strength of our approach is that we divided our problem into more focused tasks that allowed for precise model choices for each task. Alternatively, the problem of incomplete wings could be solved by using a model capable of predicting a variable number of landmarks since we have images with damaged wings containing a subset of the total number of landmarks. However, this brings various complexities to the task. We would need to include a large variety of incomplete wings, introducing sample imbalance in the training data due to varying proportions of categories of incomplete wings. In addition, from a morphometric perspective, it is standard practice to compare shapes with an equal number of landmarks. Our approach allowed us to easily create a training set and train a model with a fixed number of landmark outputs by first removing incomplete wings with a subset of landmarks. Using only complete wings avoided increased task complexity and made it directly applicable for geometric morphometrics.

There are a few limitations to the current study. Firstly, some wings with missing landmarks or other deformities may have remained in the final data set. However, these would be small in number and unlikely to bias subsequent morphometric studies. Secondly, we cannot ensure that our $R^2$ threshold method achieves perfect alignment between the images and the biological data, therefore some misalignments might still be present. This is because the threshold method is less robust in the specific case where pages have a small number of misalignments, i.e. pages with only a few misalignments are less likely to be detected since they will most likely still have a high $R^2$ value. We are more likely to pick up misalignments due to photographing mistakes that happened at the beginning of a page, since this will mean the subsequent flies will not be in the correct order, resulting in a low $R^2$. However, the sample statistics for misaligned pages indicate that the expected amount was detected, thus the majority of misalignments were found and corrected. Furthermore, some remaining misalignments were found as outliers and removed from the final data set. Thirdly, the training images may not include the full variety of shapes and sizes found in the full data set. Lastly, the incomplete wing classifier does not address some infrequent categories of incomplete wings, though the majority of incomplete wings are removed. Additionally, some of the remaining incomplete wings are identified as outliers for measured vs predicted wing length.

Providing landmarks for wings is a missing data problem and requires multiple imputation to correctly account for the extra uncertainty due to the prediction of missing values [68]. A limitation of the current study is that we provide a single set of landmarks for each wing. We caution researchers using these data for future inference that the resulting standard errors for estimates might be slightly too small. However, we feel that the practical considerations concerning the usability of the resulting landmark data set out-weigh the advantage of correctly accounting for the, likely minimal, extra uncertainty due to missing values.

We cannot know how well our model will perform when applied to different species of tsetse, or indeed different populations of G. pallidipes or G. m. morsitans. However, since we specifically aim at predicting landmarks while removing any bias toward wing shape, and since the data set used contains much of the variance we could expect (refer to Fig A of S2 Fig), we do argue that these models are unlikely to perform noticeably worse.

Several improvements can be made to increase the overall performance of these methods and their usability on more volumes of the tsetse data. Future research could improve the current incomplete wing classifier by finding a method to remove all classes of incomplete wings. Alternatively, one could develop a model that predicts landmarks for both complete and incomplete wings, paying particular attention to avoiding wing shape biases in incomplete wings. Future research could improve the data augmentation to include additional affine and elastic transformations to address inaccurate predictions for divergent shapes preventing model bias towards the mean shape. The segmentation model required a segmentation disk of

radius $R$ and used the $n_{th}$ highest pixel values to infer a final landmark location. Both these parameters were not experimented with and could be optimised to improve performance further.

To fully explore the information in the whole data set, we stress the desirability of applying the technique developed here to all wings available from all 27 volumes of data. This is, however, a major undertaking and was beyond the scope of the present study. To apply the models used in this research on other volumes of the tsetse data, one should repeat all methodologies except model training, i.e. calculating sample statistics and detecting misalignments to ensure the same level of data quality showcased in this study is achieved.

When applying these methods to other types of data, for example, the *Drosophila* data sets, one can use transfer learning to benefit from the learnt features of this study. It is important to note that the *Drosophila* data sets used by Vandaele et al. [36] and Porto et al. [37] will most likely not benefit from a deep learning approach because of the limited training data available. Therefore, we suggest a transfer learning approach using the trained network weights of our models. Loosely comparing our results with former mentioned studies, Vandaele et al. [36] reported a similar mean pixel distance error of approximately 6 pixels for images of a smaller size (1440 × 900). Porto et al. [37] reported a mean pixel distance error of 0.57%, normalising the mean pixel distance by the largest wing length. We obtained a smaller normalised pixel distance error of 0.47% for the deployed regression model using the same metric. The *Drosophila* data sets are arguably less complex than our data, which often contains image artefacts and varying wing quality that may affect the complexity of the task. Therefore we argue that successfully applying transfer learning could yield better results on these *Drosophila* data sets.

We successfully produced landmarks on an extensive data set of tsetse wing images for subsequent geometric morphometric studies. Alongside a detailed description of the methods used, we provide the trained model in the project GitHub repository [60] and the resulting landmark data from this study in data Dryad [61]. This data includes the biological recordings for each fly fully described in Table A in S1 Text, as well as the associated landmark predictions provided by this study along with all training data and annotations used to train our landmark detection models. We also provide the tsetse fly wing images from volumes 20 and 21 consisting of 14,354 images. Suggestions for studies using this data are mentioned in S3 Text under the section named *Directions for morphometric studies using these landmarks*.

## Implications for studies on the biology and control of tsetse and trypanosomiasis, and morphometric studies on wings of all insects

If, as suggested in the literature [3–10], differences in wing shape between geographically separated populations of tsetse indicate the amount of genetic flow between such populations, it is crucial to ascertain the natural variability in wing shape at any given location. The study that provided wings for the current work affords that possibility—flies being captured in 136 consecutive months, at sampling sites all within a 2km radius of Rekomitjie Research Station in the Zambezi Valley of Zimbabwe. The females captured were subject to ovarian dissection to assess their age, pregnancy state, sperm content and mortality and abortion rates [15, 17, 24–26, 29, 33]. Large subsets were also dissected to identify their trypanosome infection status [14, 16, 18, 23, 34] and were subjected to nutritional analysis to assess fat, dry weight and residual blood-meal content [13, 19–22, 27, 28, 30]. Of particular interest are two papers assessing wing length and the extent of wing fray [31, 32]. We will therefore be able to examine the relationship between wing shape and a wide variety of biological attributes of the tsetse sampled.

After a tsetse wing expands and hardens, immediately post-adult-emergence, its length does not change. [21, 22, 31]. We assume that the shape will, similarly, be independent of the

fly's age—but we will be able to check this using large samples of females where the age, and the date of birth, have been determined. Of importance will be an examination of the extent to which wing shape changes with the season and with climate variability between years across the 11 years of the study and, particularly, with the climatic conditions that a fly's mother experiences during pregnancy.

These detailed examinations of the way wing morphometrics data meshes with the underlying data set are beyond the scope of the present study, which simply aimed to establish the viability of a semi-automatic method for locating wing landmarks.

We found very good agreement between the wing lengths predicted in our study, and those measured manually using a binocular microscope in the underlying study (Fig 7). This suggests that our method should be applied to all the wings we have available ourselves, and should allow other students of tsetse populations to carry out morphometric studies on much larger samples of wings than has been feasible until now. Of more general significance is the question of whether the method is applicable to the wings of other insects, a matter that we must allow other workers to decide.

## Supporting information

**S1 Fig. Misalignment problem. Fig A.** Diagram explaining the misalignment problem and solution. The *Labeller* refers to the person who captured the images of the tsetse fly wings from the laminated page, denoted by *Page number: XXX*. The *Same image* and *Skipped image* label illustrate two types of errors that occur during labeling that result in misalignments. *Same image*, is an erroneous duplication of N2L. *Skipped image*, is an erroneously skip of image N3R, resulting in N4L being named N3R. The *Same image* error results in a misalignment of one step ahead, and one step behind for the *Skipped image* error. By analysing the correlation between predicted wing length and recorded wing length (*wlm*) using the $R^2$ metric, we are able to detect pages with misalignments. This is illustrated on the lower half of the diagram. It is easier to detect misalignments if the error happens early on in the page since more wings will be misaligned.
(TIF)

**S2 Fig. Vol. 20 and 21 motivation. Fig A.** Monthly mean wing lengths of female *G. pallidipes* sampled at Rekomitjie Research Station, Zambezi Valley Zimbabwe between September 1988 and December 1999, highlighting the results for flies from Volumes 20 and 21. The error bars show the 95% confidence intervals; where these are not visible, the bars do not extend beyond the borders of the symbol. The variation in wing length in volumes 20 and 21 covers a majority of the seasonal, and overall, variation observed with respect to wing length. As such it seems reasonable to think that the subset is sufficiently representative of the whole to support an initial attempt at developing an automatic landmark detection system.
(TIF)

**S3 Fig. Incomplete wings. Fig A.** Examples of the types of incomplete wings that appear in the data set.
(TIF)

**S4 Fig. Landmark predictions. Fig A.** Example predictions from the regression network within the 95% CI for Procrustes disparity and mean pixel distance error: green dots indicate ground truth, magenta dots indicate predictions.
(TIF)

**S1 Text. Biological recordings. Table A.** The variable names given in this table were recorded during the dissection of each respective fly. Note that these variables are only some of the variables available for each fly which are given as part of the morphometric data set published with this research.
(DOCX)

**S2 Text. Classification scores. Table A.** VGG16, parameters: 1474822. **Table B.** ResNet18, parameters: 11177025. **Table C.** Inception V3, parameters: 24346082. **Fig A.** Loss per epoch, illustrating training stability and convergence speed.
(DOCX)

**S3 Text. Data information. Table A.** A table of sample statistics for pages containing misaligned data. **Table B.** A table detailing the amount of misaligned pages found and whether they were corrected or removed from the data set.
(DOCX)

# Acknowledgments

We are grateful to C. Assumpta Nnakenyi and Vitalis K. Lagat for photographing tsetse wings.

# Author Contributions

**Conceptualization:** Dylan S. Geldenhuys, Shane Josias, Willie Brink, Jeremy Bingham, John Hargrove, Marijn C. Hazelbag.

**Data curation:** Dylan S. Geldenhuys, Willie Brink, Pietro Landi.

**Formal analysis:** Dylan S. Geldenhuys, Shane Josias, John Hargrove.

**Funding acquisition:** Cang Hui, John Hargrove, Marijn C. Hazelbag.

**Investigation:** Dylan S. Geldenhuys, Shane Josias, Pietro Landi, Jeremy Bingham, John Hargrove, Marijn C. Hazelbag.

**Methodology:** Dylan S. Geldenhuys, Shane Josias, Pietro Landi, John Hargrove, Marijn C. Hazelbag.

**Project administration:** Dylan S. Geldenhuys, Cang Hui, Pietro Landi, John Hargrove, Marijn C. Hazelbag.

**Resources:** Dylan S. Geldenhuys, Cang Hui, Pietro Landi.

**Software:** Dylan S. Geldenhuys, Shane Josias.

**Supervision:** Jeremy Bingham, John Hargrove, Marijn C. Hazelbag.

**Validation:** Dylan S. Geldenhuys, Shane Josias.

**Visualization:** Dylan S. Geldenhuys.

**Writing – original draft:** Dylan S. Geldenhuys, Shane Josias, Jeremy Bingham, John Hargrove, Marijn C. Hazelbag.

**Writing – review & editing:** Dylan S. Geldenhuys, Shane Josias, Willie Brink, Mulanga Makhubele, Cang Hui, Pietro Landi, Jeremy Bingham, John Hargrove, Marijn C. Hazelbag.

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
