## [Decision Letter · Decision Letter 0]

3 Aug 2022

Dear Mr Geldenhuys,

Thank you very much for submitting your manuscript "Deep Learning Approaches to Landmark Detection in Tsetse Wing Images" for consideration at PLOS Computational Biology.

As with all papers reviewed by the journal, your manuscript was reviewed by members of the editorial board and by several independent reviewers. In light of the reviews (below this email), we would like to invite the resubmission of a significantly-revised version that takes into account the reviewers' comments.

Please address all reviewers comments. Especially clarify the contribution and implications of the method (reviewers 1 and 2) and improve its evaluation (reviewer 3).

We cannot make any decision about publication until we have seen the revised manuscript and your response to the reviewers' comments. Your revised manuscript is also likely to be sent to reviewers for further evaluation.

Sincerely,

Virginie Uhlmann

Associate Editor

PLOS Computational Biology

Nina Fefferman

Deputy Editor

PLOS Computational Biology

Please address all reviewers comments. Especially clarify the contribution and implications of the method (reviewers 1 and 2) and improve its evaluation (reviewer 3).

Reviewer's Responses to Questions

**Comments to the Authors:**

Reviewer #1: The review is uploaded as an attachment.

Reviewer #2: This work proposes a two-step deep learning workflow for anatomical landmark detection in tsetse wings. It is evaluated on a large, original, dataset acquired and cleaned by co-authors that they plan to make available. The two main workflow steps are: 1) a classifier to identify images where landmarks are missing, 2) a landmark detection model to localize 11 interesting points (either based on regression or segmentation, with/without data augmentation). The two steps reused existing works, but the combination of the two is original. The methodology is applied on a bigger dataset and there are plans to make it available for future research.

The study is interesting and it represents quite a lot of work that could be useful for the community. But I think it would be important to better clarify the contribution, either in terms of methodology, or in terms of dataset, or both. Here are my main comments:

- For a methodological contribution, I think authors should extend their empirical evaluation to other methodologies and provide more precise empirical results. I would suggest to provide landmark prediction results without the first step (classifier) which are expected to be much lower, as a first baseline. Second, after the classification step, a second baseline could be a simple model that predicts the mean of landmark positions computed on the training set. For the segmentation approach, only a binary "disk centred at the (x,y) position of the landmark" (page 208) is evaluated while one could evaluate different heatmap generation strategies (e.g. Gaussian or exponential functions based heatmaps such as in Yeh et al., Deep learning appproach for automatic landmark detection and alignment analysis in whole-spine lateral radiographs. Scientific reports, 2021). A table with empirical results of these variants would help.

- It is also very important to clarify the empirical evaluation protocol. First, it is stated that "the study considers a data set containing 14354 pairs of images" (line 16), then authors mention they use a "one-fold cross-validation strategy" but this is a weird name for what seems to be a single independant train-validation-test split (because k-fold cross validation on 1 fold would mean dividing the full dataset in 1 fold and using 0 (k-1) fold for training). Then for the incomplete wing classifier they mention they labelled 1127 images (line 165) but evaluates it on 205 images (what is the train-validation-test split here ?). For the wing classifier, they use a full dataset of 2420 complete wings but a split of 60:20:20 (line 217). Finally, they apply their classifier to "a total of 28,708 wings" (line 310). Overall, I'm a bit lost and I have the feeling we don't have results of the full workflow on a well identified dataset but evaluations of the different steps on distinct subsets. A figure might help to understand the different subsets used as it is currently a bit difficult to draw conclusions for the whole workflow on a well identified dataset. Please clarify.

- Some choices (such as parameter values) are not well motivated: do they come from unbiased empirical evaluation or were these optimized on the test set ?. This should be clarified: Line 216 "we determined the average location of all pixels greater than or equal to the seventh-highest pixel value in the map"; Line 237 "we randomly sampled from -5,5% ... --2°,22°".

- In Discussion (line 355), authors compare their approach to other published approaches. It is a bit weak to claim that "our models perform better" because datasets are completely different. One stronger option would be to evaluate these methods on author's dataset, or to evaluate author's methodology to datasets used in these published works. Another option is to avoid such comparison or write it more carefully.

- More source code documentation would be helpful to help others reuse it.

- For a dataset contribution, authors should make it available for reviewers with more detailed description (reference 40 states "TODO" for Data Dryad) and clarify what is the "final dataset" (line 421) and line 444 "resulting landmark data": Do we get images ? How many images ? Is it 28708 ? What are the biological/technical metadata associated to them ? S1 only presents "some of the variables", what are others ? Are all these metadata directly computer-readable ? Do you provide images of the pages or individual wings ... ? Could this dataset also easily be used by other computer scientists developing other landmark detection workflow ?

Reviewer #3: In this work, the authors propose an image processing workflow for the detection of landmarks in Tsetse single-win (RGB) pictures. The number of landmarks (i.e., 11) and their positions were semantically defined. Therefore, they propose using artificial neural networks in two different ways: regression of the coordinates for each of the 11 landmarks or semantic segmentation of the landmarks. They provide an extended assessment of these approaches by using (1) the classical accuracy metrics and (2) by comparing the distances between landmarks with the distances annotated for each wing by the person who collected the data.

***Most important contributions and strengths***

- Development of a semi-automated data cleaning method to remove damaged data entries (images containing incomplete fly wings) and correct human registration errors (misaligned catalog pages). The authors estimated the expected results manually and determined that the methods performed well.

- Developed two deep-learning methods to classify tsetse fly wings based on landmarks previously established in the literature. The methods perform well and can be used to distinguish between fly populations coming from different geographic regions.

- The result assessment has been done using a final application perspective: classical accuracy metrics and morphological parameters are evaluated to assess the practicability of the method.

- Python code was made publicly available, is annotated in a comprehensive manner, and is reusable in Jupyter Notebooks. The authors used a publicly available annotated dataset.

***Weak points***

- Semi-automated data cleaning step to remove images containing incomplete fly wings is based on the detection of 2 out of 11 landmarks. The authors estimate these to comprise ~77% of the total number of incomplete wings, leaving behind an estimated ~23%. This can be improved. Likewise, S3 section shows that the classification models have mostly a perfect performance. Does this mean that these models are only capable of detecting 77% of the damaged wings?

- The catalog page alignment method is less robust at detecting pages with small misalignments, especially if they do not occur at the beginning of a page.

- The authors propose the application of standard deep-learning techniques to build a workflow. Therefore, the current work lacks methodological novelty.

***Major comments***

Abstract: The information is very complete but I would strongly suggest the authors reorganise it. For example, the information about the number of images could be given together with the size of the images at the end, when results are mentioned. When the authors say “fully convolutional networks”, they may be referring to “fully connected convolutional neural network”. The current work does not propose a new approach to deal with the unbalance between the background and foreground (weighted loss functions, focal loss, or specific sampling) but it uses the common dice loss function, so I strongly recommend removing the statement about “data-centric approach paying attention to consistent labelling”. Moreover, data augmentation is a common procedure when training convolutional neural networks, so I would not highlight it in the abstract. The following sentence could be rephrased as the authors do not provide data but a reusable code (“The resulting landmark data set was provided for future morphometric analysis.”). Moreover, the current work does not provide real units of size (centimeters or millimeters) for the pixels, which could be confusing if used for further biological research.

Figure 1: The authors say: “The image also contains a scale that can be useful for placing later errors into context” Are the authors referring to the 50 pixels scale bar in the image? If so, this scale bar is virtually added so it does not provide additional value to the framework. Also, the method is proposed to measure wing morphology, so the authors should strongly consider recovering the original pixel size of the images.

Figure 2: It has poor information to become one single figure. This is a potential proposal to consider: As the authors show the regression approach in this figure, they could merge the information from Fig4 and also show the segmentation approach already shown in Fig3.

Still, if the authors decide to keep Figs 3 and 4 independently, then Figs 1 and 2 could be merged to provide a summary of the proposal, and Figs 3 and 4 should be more detailed (e.g., number and size of convolutional layers, input and output sizes, whether skip connections exist).

The page misalignment problem and its proposed solution is unclear in the text. Adding visual examples or representing this information graphically could be a useful way to make it easier for readers to understand it.

The proposed method performs an image transformation to have the wings always oriented in the same manner. On the other side, the network sees the entire wing in one shot (the input image) and each layer of the output (both for regression and segmentation) belongs to a semantic class (e.g. 5th layer infers landmark number 5). Therefore, the necessity to reorient the wings is not clear. Additionally, should this transformation be done manually one by one as a preprocessing step?

The authors adapt the shape of the pictures to the input size of the convolutional neural networks. Usually, they downsample them. How does this affect the disks used for the landmarks in the segmentation approach?

* Evaluation of the Landmark detection:*

While the authors provide a nice assessment of the method using morphological parameters and metrics, knowing whether errors can be negligible or not could be improved by knowing the pixel size.

Figure 5: It is used to compare both approaches but it is difficult to see if there exists any practical difference in the performance between both (except for the fact that the regression model demands fewer resources). A potential improvement would be to show this data on a logarithmic scale. Additionally, the segmentation model seems to provide more stable results rather than the regression one.

Method comparison: the authors decide to evaluate the accuracy of the models for each landmark. However, because the 11 landmarks depend on the entire wing (they are not independent), it makes more sense to evaluate how well the method performs for each independent pair of wings of each Tsetse individual, or if not possible, for each wing.

In Page 11, the authors argue why they think the regression model is more stable than the segmentation. This statement sounds too strong while the evidence from Figure 6 and 5 are not that clear: The segmentation model has way smaller averaged errors than the regression model (approximately 4 pixels). Knowing that a landmark is represented by one single pixel, it is critical to know what is the pixel size to really consider this amount negligible or not. Moreover, the fact that the errors are not correlated with the Procrustes disparity could be interpreted that the method is inferring the landmark positions regardless of the true shape of the wing, so it is agnostic to the wing model used for Procrustes.

The authors propose the incomplete wing classifier as a novel application of the task. The models tested have all of them, mostly a perfect performance. The differences are due to the model initialisation or the model instance, which has a random component. In such a case, it is quite interesting to know which model requires larger resources, has fewer parameters, or converges faster.

***Minor comments***

For the segmentation methods, the authors calculate a disk around the landmark. How is the diameter of this disc calculated? For detection, it is common to use Gaussian disks so the training and loss functions are not that unstable in presence of outliers or small deviations with respect to the natural intersection of veins in the wing.

The section “Directions for morphometric studies using these landmarks” is slightly redundant with the content of the main text. It could be more useful to provide a guideline of what is the exact data format that the code can process, what are the considerations to have when taking pictures to the wings so the workflow can be reused, or more tips in terms of exploiting the current method.

Consider correcting the color palette for colorblindness—particularly Fig 7, 8, and S4.1, which give ambiguous information for certain types of colorblindness. Use an online tool such as https://colororacle.org/ to simulate colorblindness, and refer to https://www.nature.com/articles/d41586-021-02696-z for palette choice. Remember to update legends after changes are made.

Training images for the wing classification model all come from a subset of the entire dataset corresponding to a specific year (1995). The authors do not address the possibility of the emergence of new variants in later years, to which the trained model would be agnostic.

*** Small changes and typos***

Please substitute all the Unet by UNet or U-Net.

Confidence intervals are sometimes written as CI and others as confidence intervals. Please, be consistent along the manuscript.

Figure 7: in the y axis it is said between landmark 0 and 5 and in the caption, it is said 1 and 6. Please, correct it.

Table 1: what are the metrics of wlm? this applies to Figure 7.

Line 18: “The data form a subset of a larger collection of more than 200,000 pairs of wings” → “The data belongs to a larger collection of more than …”

Line 19: “On each intact wing are 11 landmarks, defined…” → On each intact wing 11 natural landmarks are identified as the points of …

Line 20 – The link for Fig 1 directs the reader to the bottom of the page (possibly linked to the page number) instead of the actual figure.

Line 30: “Although detecting incomplete wings WAS not of concern in this study, their methods may be good candidates for our task since both tasks aim at wing classification.” To which tasks are the authors referring with “both tasks aim at wing classification”?

Line 31 – Replace “Although detecting incomplete wings were not of concern in this study, their methods may be a good candidate for our task since both tasks aim at wing classification.” to “Although detecting incomplete wings was not of concern in this study, the methods used may be a good candidate for our task since it also aims at wing classification.”

Line 38 – The term “Convolutional neural network” is used here for the first time and repeated throughout the manuscript. Consider defining an abbreviation (e.g., CNN) here and use that abbreviation in subsequent mentions.

Line 39: “the best bottleneck features”: for which task?

Line 44: “fully CONNECTED convolutional neural networks.”

Line 46 – The term “Fully convolutional” is used in its hyphenated form in line 44. Check the consistency by using the same form throughout the manuscript.

Line 51 – Problem in citations. “[?, 20, 21]”.

Line 82 – The link to “S1 Table” directs the reader to page 7 instead of the actual table.

Line 83 – Link to Table 1 directs the reader to the bottom of the page (possibly linked to the page number) instead of the actual table.

Line 87 – Replace “The distance between landmarks 1 and 6 (refer to Fig 1) was used as this standard measure, converted to a length (wlm) in mm by allowing for magnification differences between microscopes.” by “The distance between landmarks 1 and 6 (refer to Fig 1) was used as this standard measure, converted to a length (wlm) in millimeters, allowing for magnification differences between microscopes.”

Line 96 – “These volumes are also considered sufficiently large to incorporate much of the variation in all volumes and are suitable for an initial attempt at applying an automatic landmark detection system.” Please provide a brief explanation for the basis of this assumption. Volumes 20 and 21 correspond to only ~7% of the total dataset, which may lead to doubts. Is it possible that new variants emerge throughout the years and consequently their weight is not taken into account from here on?

Line 114 – The link for the S2 Table directs the reader to the end of page 17 and not the table.

Line 119: “The sample statistics showed that 13% +- 4.63% of wings are incomplete” How was this metric calculated? In particular the standard deviation.

Line 121 – Replace “The majority of incomplete wings were missing landmark 4 or 6, making up 10% ± 4.13% of the data, and comprises on average 77% of all incomplete wings.” by “The majority of incomplete wings were missing landmark 4 or 6, making up 10% ± 4.13% of the data, comprising on average 77% of all incomplete wings.”

Line 122 – Insert quotation marks or use italics in the terms “texitlmkr” and “textitlmkl”.

Line 131 – “We then removed incomplete wings using a classification approach. Since most incomplete wings are missing landmark 4 or 6, we trained a classifier to identify wings missing these landmarks.”. Training the classifier to detect other landmarks should improve the performance of this task. Assuming the algorithm correctly classifies 100% of the wings lacking landmarks 4 and 6, a remainder of ~23% of incomplete wings are not classified accordingly. Nonetheless, this question is adequately addressed by the authors throughout the manuscript.

Line 154 – Correct section title “incomplete wing classifier” to “Incomplete wing classifier”.

Line 158 - Replace “Since manually finding training samples is time-consuming, we aided the process by using the biological data set (variables lmkr and lmkl) and then use visual inspection to filter a set of training images.” by “Since manually finding training samples is time-consuming, we aided the process by using the biological data set (variables lmkr and lmkl) and then used visual inspection to filter a set of training images.”

Line 162 – Replace “We removed bad types of examples that do not clearly fall into either class” by “We removed examples that do not clearly fall into either class”.

Line 167 - “We compared three modern computer vision models with transfer learning. These models are ResNet18 [26], Inception [27], and VGG16 [28] with batch normalisation.”. Please provide a brief explanation for the basis of choosing these models specifically.

Line 279: “a mean average error (MAE) OF the Euclidean pixel distance”

Table 2 – A plot instead of a table could make these results easier to visualise.

Line 361 – “Drosophila data sets are arguably less complex than our data, which often contains image artefacts and varying wing quality which may affect the complexity of the task.” by “Drosophila data sets are arguably less complex than our data, which often contains image artefacts and varying wing quality that may affect the complexity of the task.”

S5: “Alterations to volumes 20 and 21” – Missing spaces between numbers in the “12 pages corrected” line.

**Have the authors made all data and (if applicable) computational code underlying the findings in their manuscript fully available?**

Reviewer #1: **No: **The code is availabe but suffers from errors. No repository description is available. Code would benefit from DOI. It is unclear whether the data are from a previous study by Hargrove et al. and the according reference to Dryad is incomplete “Data Dryad Tsetse landmarks for volumes 20 and 21 TODO”.

Reviewer #2: **No: **Code: yes on Github (python notebook) ; Data: No reference 40 states "TODO" for Data Dryad

Reviewer #3: Yes

PLOS authors have the option to publish the peer review history of their article (what does this mean?). If published, this will include your full peer review and any attached files.

Reviewer #1: **Yes: **Elisabeth Kugler

Reviewer #2: No

Reviewer #3: **Yes: **Estibaliz Gómez de Mariscal
---

## [Decision Letter · Decision Letter 1]

1 Feb 2023

Dear Geldenhuys,

Thank you very much for submitting your manuscript "Deep Learning Approaches to Landmark Detection in Tsetse Wing Images" for consideration at PLOS Computational Biology.

As with all papers reviewed by the journal, your manuscript was reviewed by members of the editorial board and by several independent reviewers. In light of the reviews (below this email), we would like to invite the resubmission of a significantly-revised version that takes into account the reviewers' comments.

Although Reviewers suggested a minor revision, two of the issues raised are absolutely critical and must be addressed before we consider this submission for publication: the first one relates to reproducibility, as highlighted by Reviewers 1 and 3, and the second one to baseline comparison, as raised by Reviewer 2. Addressing these points may not necessarily require major changes to the text of the manuscript as they revolve around numerical experiment, data, and code, but must nevertheless be done.

We cannot make any decision about publication until we have seen the revised manuscript and your response to the reviewers' comments. Your revised manuscript is also likely to be sent to reviewers for further evaluation.

Sincerely,

Virginie Uhlmann

Academic Editor

PLOS Computational Biology

Nina Fefferman

Section Editor

PLOS Computational Biology

Although Reviewers suggested a minor revision, two of the issues raised are absolutely critical and must be addressed before we consider this submission for publication: the first one relates to reproducibility, as highlighted by Reviewers 1 and 3, and the second one to baseline comparison, as raised by Reviewer 2. Addressing these points may not necessarily require major changes to the manuscript, but must nevertheless be done.

Reviewer's Responses to Questions

**Comments to the Authors:**

Reviewer #1: In this revision, Geldenhuys et al. present a significantly improved manuscript on a two-tier method using deep learning architectures to classify images and make accurate landmark predictions of tsetse fly wings.

A lot of the work in the revision has gone into the introduction, which now sufficiently highlights the study rationale on wing morphometry as well as provides more context for deep learning. Also, the discussion has significantly expanded on application to other data. Overall, the manuscript has improved and now also states the study limitations and potential future work (e.g. hyperparameters and comparisons to other studies).

Major comments:

• While the code has now a DOI and is referenced appropriately, there is still no readme in the GitHub repository and no code changes in 16 months. Thus, usability and reproducibility of this work are severely impacted (as raised by multiple reviewers). Making code available is great, but it only makes sense when others can use it (i.e. executable, documentation, etc).

• The authors now briefly touch on the point of “signal-to-noise ratio and data quality”, but again address this very superficially with a computational mindset. My initial point was more with respect to the question of “How good do your data need to be?”

Minor comments:

• Line 87 - “.” is missing

• Figure 2 – at the end of the paragraph “.” Is in the next line

• Data Description section: “Nonetheless, to further explore the variation in the whole data set, we stress the desirability of applying the technique developed here to all wings available from all 27 volumes of data. This is, however, a major undertaking and was beyond the scope of the present study.” – This paragraph needs to go into discussion.

• line 335 – typo “squared” should be “squared”

Reviewer #2: Authors have rewritten abstract and introduction to clarify the contributions of this work, and improved the description of evaluation protocols, and discussion. Overall the quality of the paper has been improved.

However, I still have minor comments:

1) I do not understand why authors did not perform the suggested, very basic, evaluation of baseline (mean position prediction for landmark detection, on the same data and protocol as line 230) without giving a convincing argument about not implementing it. Indeed it should take less than one day on the tetse_data.csv: for each landmark compute the mean coordinates values on the learning set, use these mean values as predictor, compute metrics on the test-set, report them similarly as you did for deep learning models. I still believe such a baseline is important when releasing a new dataset. This basic evaluation could strengthen the interest of using their deep learning approaches and let readers better assess how difficult is the task. Instead, authors only replied "The suggestion for trying a model that predicts the mean of landmark positions from the training set is a neat way of comparing how much better the models perform compared to the most straightforward solution. We suggest that the models we present are baseline models, i.e. a good starting point for further development".

2) Their answer concerning some parameter value choices (Section training implementation, data augmentation) is OK, but this could be translated to an additonal sentence in the main text stating that for these parameters "we chose them based on a visual inspection of what degree of augmentation seemed reasonable" (line +/- 280).

3) I don't think the abstract and author's summary should mention "The methods we have developed should apply to studying the wings of any insect species" (abstract) neither "Our method applies to the study of the wings of any insect species" (author's summary) as it has not been tested (as recognized later in main text line 570-571). Moreover, although the source code is provided, it is in a state that will not facilitate direct reuse on other data as it is provided as Jupyter Notebook without library versioning (a pip requirements file would be welcome), with specific filenames explicitly mentioned in the source code, and with very little documentation. So I think applying the methodology on other data would not be straightforward.

4) "sqaured" line 335 -> "squared"

Reviewer #3: I thank the authors Dylan Geldenhuys et al., for taking the time and the efforts to kindly answer all the highlights made in the previous review and considering them to update the current work.

Most of the comments made from this side were addressed. However, there are still certain concerns that should be highlighted:

Section Weak points:

- "We agree that despite our best efforts, we might not have caught all misalignments. As the reviewer mentions, the alignment method is less robust in the specific case where pages have a small number of misalignments; hence, we are likely to detect the majority of misalignments. Furthermore, our sample statistics (lines 378-382) suggest that we could detect the expected amount of misaligned pages."

I agree with the authors that having a method that performs accurately for the current task is a good solution. However, I would suggest including further information about the algorithm's limitations in the discussion so other researchers can take it into account

Section Major comments:

- Figure 2: It is a supportive graphical information to understand the pipeline. If possible, I would suggest the authors switching the order of the tasks (segmentation and classification) to align with the order of the tiers.

- Wing reorientation: While it is true that this task is not a milestone of the proposed pipeline, the following statement could be slightly strong. This is particularly important due to the lack of numerical comparison between the non- and reorientation approaches: "The orientation of features (veins) for each semantic class would only be consistent if we trained it by first converting all images to face the same direction. If we apply the regression network on an image facing the wrong direction, it won't perform very well." CNN can learn to be invariant to translation/rotation quite easily and it is indeed a typical feature when analysing bio-images. Namely, in this case the network sees the entire wing so it should be even easier for the CNN to identify all the landmarks. For this reason, without strong evidences, I would recommend the authors to smooth such statements.

- The authors use an online calculator for the sample size estimator (https://www.calculator.net/sample-size-calculator.html). It would be recomendable including this information in the manuscript to ensure reproducibility.

**Have the authors made all data and (if applicable) computational code underlying the findings in their manuscript fully available?**

Reviewer #1: **No: **The current code state does not allow for it to be used by others. - There is insufficient documentation, errors, and no readme. The authors state they will make a readme available, but no repository changes have been made in months. I would not support publication if this issue is not sorted.

Reviewer #2: Yes

Reviewer #3: **No: **The following link does not give access to the data as the link is empty: "The training data as well as the final landmark data can be found in the Data Dryad repository " ext-link-type="uri" xlink:type="simple">https://orcid.org/0000-0002-7702-4831"

PLOS authors have the option to publish the peer review history of their article (what does this mean?). If published, this will include your full peer review and any attached files.

Reviewer #1: **Yes: **Elisabeth Kugler

Reviewer #2: No

Reviewer #3: **Yes: **Estibaliz Gómez de Mariscal

Figure Files:

Data Requirements:

Reproducibility:

To enhance the reproducibility of your results, we recommend that you deposit your laboratory protocols in protocols.io, where a protocol can be assigned its own identifier (DOI) such that it can be cited independently in the future. Additionally, PLOS ONE offers an option to publish peer-reviewed clinical study protocols. Read more information on sharing protocols at https://plos.org/protocols?utm_medium=editorial-emailutm_source=authorlettersutm_campaign=protocols

---

## [Decision Letter · Decision Letter 2]

17 May 2023

Dear Geldenhuys,

We are pleased to inform you that your manuscript 'Deep Learning Approaches to Landmark Detection in Tsetse Wing Images' has been provisionally accepted for publication in PLOS Computational Biology.

Best regards,

Virginie Uhlmann

Academic Editor

PLOS Computational Biology

Nina Fefferman

Section Editor

PLOS Computational Biology

Reviewer's Responses to Questions

**Comments to the Authors:**

Reviewer #1: The authors have increased the reproducibility of their work by significant work on the code repository. The GitHub repository now contains updated notebooks, documentation, and information towards example data.

I think the manuscript has greatly improved during the revision and advice the current version to be accepted.

Reviewer #2: Authors have tone down their reuse claim, documented the source code, and provided the baseline results. I have no further comments.

Reviewer #3: I thank the authors for addressing the comments made in the previous review and including further comments in the manuscript. The authors also corrected the links in the GitHub repository and now the data and models are publicly available.

**Have the authors made all data and (if applicable) computational code underlying the findings in their manuscript fully available?**

Reviewer #1: Yes

Reviewer #2: Yes

Reviewer #3: Yes

PLOS authors have the option to publish the peer review history of their article (what does this mean?). If published, this will include your full peer review and any attached files.

Reviewer #1: **Yes: **Elisabeth Kugler

Reviewer #2: No

Reviewer #3: **Yes: **Estibaliz Gómez de Mariscal, Afonso Mendes

---

## [Editor Report · Acceptance letter]

22 Jun 2023

PCOMPBIOL-D-22-00611R2 

Deep Learning Approaches to Landmark Detection in Tsetse Wing Images

Dear Dr Geldenhuys,

I am pleased to inform you that your manuscript has been formally accepted for publication in PLOS Computational Biology. Your manuscript is now with our production department and you will be notified of the publication date in due course.

With kind regards,

Zsofi Zombor
